# Effects of aquatic exercise on arterial stiffness and endothelial function in adults: A systematic review and meta-analyses

Emily Dunlap[1]*, Yanbing Zhou[1], Manny M.Y. Kwok[2], Billy C.L. So[2], Hirofumi Tanaka[1]

**1** Department of Kinesiology and Health Education, The University of Texas at Austin, Austin, Texas, United States of America, **2** Department of Rehabilitation Sciences, The Hong Kong Polytechnic University, Kowloon, Hong Kong

\* edunlap@utexas.edu

## Abstract

### Objective

To evaluate the effects of aquatic exercise compared with non-exercise controls and land-based exercise on arterial stiffness and endothelial function.

### Design

Systematic review and meta-analyses of randomized controlled trials assessed using the Cochrane risk-of-bias tool and Grading of Recommendations Assessment, Development and Evaluation.

### Data sources

PubMed/MEDLINE, CINAHL Plus, SPORTDiscus, and reference lists, searched from database inception to April 16, 2025.

### Eligibility criteria

Studies evaluating chronic aquatic exercise (multi-session interventions) compared with land-based exercise or non-exercise comparison groups in adults, measuring arterial stiffness via pulse wave velocity (PWV) or endothelial function via flow-mediated dilation (FMD).

### Results

This review includes 18 randomized controlled trials with 845 participants (mean age 65±7 years). Studies compared aquatic exercise with non-exercise controls (8 studies), land-based exercise (6 studies), or both (4 studies). Exercise sessions averaged 50 minutes, 3 times weekly for 11 weeks. Most studies (17 out of 18) implemented moderate-to-vigorous intensity protocols. Aquatic exercise resulted in improvements

**Data availability statement:** All data underlying the research findings described in our manuscript are within the manuscript or in the supporting information.

**Funding:** The author(s) received no specific funding for this work.

**Competing interests:** The authors have declared that no competing interests exist.

in arterial stiffness compared with non-exercise controls (7 studies; SMD = −2.37, 95% CI: −4.46 to −0.29; $I^2$ = 98%: low certainty), with most evidence reflecting systemic and peripheral PWV. Changes in arterial stiffness did not differ from those observed after land-based exercise (6 studies; SMD = −0.07, 95% CI: −0.34 to 0.20; $I^2$ = 0%, moderate certainty). For endothelial function, aquatic exercise may improve outcomes versus non-exercise controls (6 studies; SMD = 0.91, 95% CI: 0.39 to 1.43; $I^2$ = 68%; low certainty) and may lead to greater improvements than land-based exercise (7 studies; SMD = 0.55, 95% CI: 0.05 to 1.06; $I^2$ = 75%; low certainty).

### Conclusion

Aquatic exercise improves systemic and peripheral arterial stiffness as well as endothelial function compared with non-exercising controls. Changes in arterial stiffness do not differ from those observed after land-based exercise. Aquatic exercise may provide greater improvement in endothelial function than land-based exercise, though this is supported by low-certainty evidence, and substantial heterogeneity limits confidence in the generalizability of this finding.

### PROSPERO registration

CRD42025642087.

### Introduction

Vascular dysfunction, characterized by arterial stiffening and endothelial impairment, develops with advancing age and physical inactivity, and contributes to cardiovascular disease (CVD) [1]. To evaluate these vascular function indicators, pulse wave velocity (PWV) and flow-mediated dilation (FMD) serve as key non-invasive diagnostic tools. Meta-analytical evidence highlights that a 1% improvement in FMD correlates with an 8–16% reduction in fatal or non-fatal CVD risks and all-cause mortality, with even more pronounced benefits in individuals with pre-existing CVD [2]. Similarly, a 1 m/s decrease in PWV corresponds to a 12–15% decline in cardiovascular events, 13–15% lower CVD mortality, and a 6–15% reduction in all-cause mortality [3].

Exercise training plays a vital role in preventing CVD and reducing mortality risk. Aerobic exercise, in particular, has been shown to improve vascular function by reducing arterial stiffness in middle-aged and older populations [4]. These vascular adaptations are driven by a reduction in α-adrenergic receptor-mediated vascular tone [5] as well as transient surges in vascular shear stress during exercise, initiating improvements in endothelial function [6,7]. This evidence suggests that strategically modulating sympathetic vasoconstrictor tone and shear stress during exercise could augment and optimize vascular benefits.

Land-based exercise poses significant barriers for individuals with vascular dysfunction, including symptoms such as immobility and fatigue, as well as functional

limitations [8]. Joint and muscle injuries associated with land-based activities may further hinder exercise adherence [9]. However, aquatic exercise could help reduce these barriers by providing a low-impact environment, potentially promoting better long-term adherence [10]. Compared with land-based exercise, aquatic exercise amplifies vascular shear stress [11], partly due to hydrostatic pressure during immersion, which enhances venous return from the lower limbs. This elevates cardiac venous return, ventricular volume, and ultimately cardiac output [12]. Additionally, water immersion decreases sympathetic nervous system activity [13], with temperature-dependent effects further influencing shear stress dynamics in peripheral arteries [14].

Aquatic exercise has emerged as a promising alternative for individuals with joint pain or mobility challenges, leveraging water buoyancy to reduce mechanical stress on joints while facilitating physical activity due to characteristics that are exclusive from the land environment [15]. The unique hemodynamic and thermal properties of water immersion position it as a compelling intervention for enhancing vascular function [14]. Existing research presents conflicting results regarding the efficacy of aquatic exercise compared with non-exercise groups or land-based interventions in reducing arterial stiffness or improving endothelial function. While some studies have found improvements in FMD and PWV following aquatic exercises such as swimming [16,17] and aquatic walking [18], others have reported no significant differences compared with land-based or non-exercise control groups [19,20]. Unfortunately, the effects of aquatic exercise on arterial stiffness and endothelial function have not yet to be systematically reviewed. The primary aim of this systematic review with meta-analyses is to compare the effects of chronic aquatic exercise, land-based exercise, and non-exercise control conditions on arterial stiffness and endothelial function, with the goal of clarifying their relative contributions to vascular function improvement in adults.

## Methods

All procedures undertaken in the present review adhered to the reporting standards established in the Preferred Reporting Items for Systematic Reviews and Meta-Analyses statement (PRISMA) [21,22] and were prospectively registered with the International Prospective Register of Systematic Reviews (PROSPERO: CRD42025642087). The University of Texas at Austin IRB determined that the proposed activity is not research involving human subjects as defined by DHHS and FDA regulations. This study involves a meta-analysis of published research studies available through academic and scientific databases. It does not involve the collection or use of private, identifiable data from individual participants, so no human subjects are involved. Equity, diversity, and inclusion were considered in study selection and data extraction.

### Search strategy and study selection

A systematic search was conducted in PubMed, CINAHL, and SPORTDiscus from inception to April 16, 2025. The search strategy is presented in the supporting information (S1 File). In addition to the database search, a manual search of reference lists in relevant articles was undertaken to identify potentially eligible studies for inclusion. Following the removal of duplicates, two of the three reviewers (ED, MK, and YZ) independently screened titles and abstracts using the web-based systematic review software Rayyan [23]. The screening process was blinded, with reviewers unable to see each other's decisions until completion. Conflicts were resolved through discussion between reviewers, with a third reviewer consulted when consensus could not be reached. When abstracts did not provide sufficient information, they were selected for full-text evaluation. Full-text articles meeting the criteria were retrieved and read independently by the reviewers and assessed for study inclusion.

### Eligibility criteria

We included controlled trials evaluating an aquatic exercise intervention compared with non-exercise and/or land-based exercise control groups. Primary outcomes were arterial stiffness measured by PWV and endothelial function measured by FMD. The inclusion criteria were: (1) adults (≥18 years of age), including both healthy individuals and those with

chronic health conditions; (2) prospective controlled trials; (3) chronic aquatic exercise involving multi-session physical activity performed in water (e.g., swimming, water walking, aquarobics); (4) comparison of either land-based exercise and/or non-exercise group; and (5) outcome measures included PWV and/or FMD. Exclusion criteria included: (1) pool water temperature greater than 36°C or lower than 20°C; (2) animal models; (3) water immersion only without active exercise; (4) scuba or diving reflex studies; (5) single-session aquatic exercise (common in crossover trials) and (6) non-peer-reviewed articles.

## Data extraction

Data extraction was independently and separately performed by two of the three authors (ED, MK, and YZ), with disagreement resolved by a third author. Data extraction included sample size, population, age, sex, group details (exercise type, length, frequency, duration, and intensity), baseline and post-intervention outcomes measurements of interest (PWV and/or FMD), water temperature and depth, exercise adherence and adverse effects. Any outcomes of interest not reported as means and SDs were converted with the following formulas:

SEM converted to SD: [24]

$$SD = SEM \times \sqrt{n}$$

Median and 25–75th percentiles converted to mean and SD: [25]

$$Mean = (Q1 + median + Q3)/3$$

$$SD = \sqrt{\left(\left((Q1 - mean)^2 + (median - mean)^2 + (Q3 - mean)^2\right)/3\right)}$$

## Data analyses

All statistical analyses and forest plots were generated using Review Manager Web (RevMan Web, Version 5.4, The Cochrane Collaboration, 2020). Meta-analyses examined aquatic exercise effects on arterial stiffness (measured with PWV) and endothelial function (measured with FMD), with separate analyses comparing aquatic exercise versus non-exercise controls and versus land-based exercise.

We calculated standardized mean differences (SMDs) based on change from baseline measures with 95% confidence intervals. SMDs were used rather than mean differences to account for methodological variability across studies. PWV was measured at different arterial segments (central, peripheral, or systemic), and while most studies assessed brachial FMD, one study measured popliteal FMD. Using SMDs allowed standardization of effect sizes across these different measurement sites. When SDs of the change scores were not reported, we estimated them using a conversion formula: $SD\_diff = \sqrt{(SD\_pre^2 + SD\_post^2 - 2r \times SD\_pre \times SD\_post)}$, assuming a correlation coefficient (r) of 0.5 between pre- and post-intervention measures [26]. When studies reported multiple PWV measurements, we applied specific data selection criteria. Where right and left side brachial-ankle PWV were reported separately [27], we calculated the average of the two measurements and computed the pooled standard deviation using the formula: $SD\_pooled = \sqrt{[(SD_1^2 + SD_2^2)/2]}$. In cases where both femoral-ankle PWV and brachial-ankle PWV were available [28], brachial-ankle PWV data were used as it is a more established measure of arterial stiffness [29]. When PWV measures for both paretic and non-paretic legs were reported in adults after stroke [30], we selected the non-paretic leg data to minimize the potential confounding effects of hemiparesis on vascular measurements. Random-effects meta-analysis models were used because we anticipated true variation in effect sizes across studies due to differences in populations, intervention protocols, and

measurement techniques. Models were conducted using the restricted maximum-likelihood (REML) method for estimating between-study variance (Tau$^2$), with confidence intervals calculated using the Wald-type method. When heterogeneity was moderate or higher (>50% I$^2$), subgroup analyses were conducted to explore potential sources of heterogeneity. These subgroups were not pre-specified in the PROSPERO registration but were conducted post hoc to explore potential sources of heterogeneity identified in the primary analyses. To account for uncertainty in between-study variance, 95% prediction intervals were computed using the Hartung–Knapp–Sidik–Jonkman method.

## Quality of evidence assessment

**Risk of bias assessment.** Two reviewers (ED and YZ) independently assessed risk of bias for each included study using the Cochrane risk of bias tool for randomized trials (RoB 2) [31]. The assessment was conducted using the algorithm and signalling questions provided in the RoB 2 tool [32]. The RoB 2 tool evaluates five domains: bias arising from the randomization process, bias due to deviations from intended interventions, bias due to missing outcome data, bias in measurement of the outcome, and bias in selection of the reported result. Each domain was rated as 'low risk,' 'some concerns,' or 'high risk,' with an overall risk of bias judgment determined by the highest risk rating across all domains. Disagreements were resolved through discussion.

**Certainty of evidence assessment.** The certainty of evidence for each vascular outcome was assessed using the Grading of Recommendations Assessment, Development and Evaluation (GRADE) approach. GRADE assessments were conducted using GRADEpro GDT software (McMaster University and Evidence Prime, Hamilton, Canada) integrated within Review Manager Web (RevMan Web, Version 5.4, The Cochrane Collaboration, 2020). Evidence certainty was evaluated based on five factors that may decrease confidence (risk of bias, inconsistency, indirectness, imprecision, and publication bias) and three factors that may increase confidence (large magnitude of effect, dose-response gradient, and effect of plausible confounding). Evidence was rated as high, moderate, low, or very low certainty.

## Results

### Study selection and characteristics of included studies

The comprehensive search across databases and other sources yielded 4,386 results as shown in Fig 1. After duplicates were removed, the initial screening with title and abstract identified 50 reports targeted for full retrieval. One report could not be retrieved because the full text was unavailable. Of the remaining 49, full-text review led to the exclusion of 31 reports. Reasons for the exclusion are detailed in Fig 1. Finally, 18 randomized controlled trials were included for the meta-analyses. Eight studies compared aquatic exercise with non-exercise controls only, while six studies compared aquatic exercise with land-based exercise [16,17,28,30,33,34], and four studies compared aquatic exercise with both [20,35–37].

The characteristics of the 18 included studies are shown in Table 1. A total of 845 participants were involved in the review with an average age of 65±7 years (range 52–76 years). Sample sizes ranged from 18 to 100 participants. The studies represented a variety of adult populations: six focused on generally healthy adults, while twelve involved participants with specific conditions, including type 2 diabetes (n=3), peripheral artery disease (n=2), coronary artery disease (n=2), hypertension (n=2), osteoarthritis (n=1), obesity (n=1), and subacute stoke (n=1). Aquatic exercise interventions varied across studies. Six studies implemented water walking protocols [18,20,27,28,30,39], while another six used combined walking and aerobic exercises [19,35–37,40,41]. Four studies used swimming [16,17,38,42], while individual studies examined aquatic bicycling [33] and high-intensity aquatic running intervals [34].

The dosage of the aquatic exercise ranged from 20 to 60 minutes per session, two to five sessions per week, for two to 24 weeks. The average dosage from all studies was 50 minutes, three times per week, for 11 weeks. The intensity of almost all studies were reported as a percentage of heart rate with seven studies reporting intensity in an range of 30% to

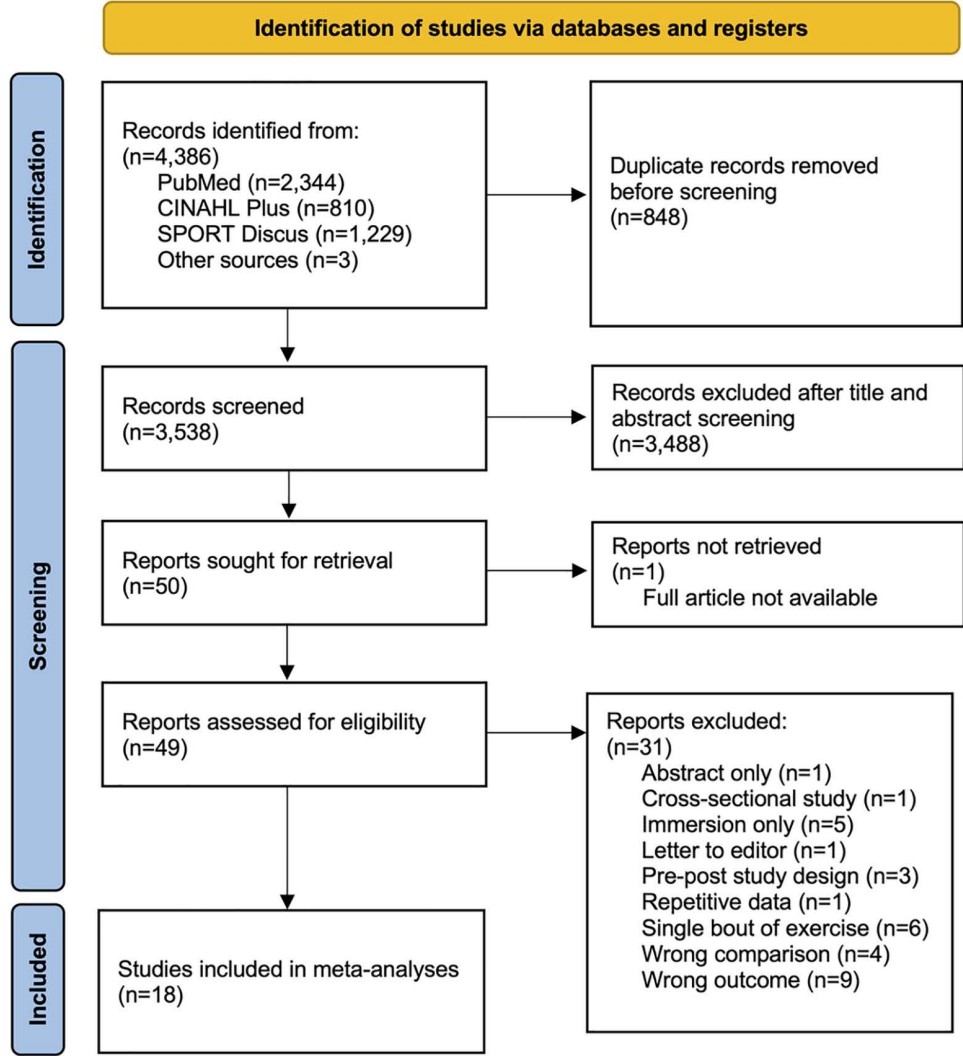

**Fig 1. Study selection flow diagram.**

75% (average 41% to 68%) of heart rate reserve [16,20,34,35,39,41] and nine studies listing an intensity range of 30% to 85% (average 52%−75%) of maximal heart rate [17,18,28,30,33,36,37,40,42]. This would categorize the intensity of these aquatic programs as moderate to vigorous exercise according to the American College of Sports Medicine [43].

The pool water temperature reported in 14 studies averaged 31°C. The water temperature in seven studies was set between 28 and 30°C [16,18–20,27,34,35], which is considered an appropriate temperature for community pools [44], while seven studies indicated a water temperature between 30 and 35°C [28,30,33,36,37,39,40], which is a common temperature for therapy pools [44]. The depth of the water varied with two studies as shallow as knee-to-waist level [30,33] and others listed either waist-to-chest level or from 1.2 m to 1.5 m [18,20,27,28,34–37,39].

The vast majority of studies evaluating endothelial function measured FMD of the brachial artery with an occlusion duration ranging from 3–5 minutes [16,17,20,34,36–40], while one study used popliteal FMD with a 3-minute occlusion [33]. Four studies evaluating arterial stiffness utilized peripheral pathways of carotid-brachial [19], carotid-radial [42], and femoral-ankle [18]. Three studies assessed central arterial stiffness using carotid-femoral PWV [16,34,41], while the

**Table 1. Characteristics of included studies.**

| Studies | Sample | Aquatic Exercise Group | | | | | | | Comparison Group | | | | | |
|---|---|---|---|---|---|---|---|---|---|---|---|---|---|---|
| | | Age years ±SD | n total % male | Type | Pool temperature depth | Dosage | PWV type baseline ±SD (m/s) | FMD type baseline ±SD (%) | Age years ±SD | n total % male | Type | Dosage | PWV type baseline ±SD (m/s) | FMD type baseline ±SD (%) |
| **Aquatic vs Non-Exercise Control Group** | | | | | | | | | | | | | | |
| **Ha et al. 2018** [19] | Older women | 74 ±4 | 11 0% | aquarobics | 27°C NR depth | 50min 3 x/week 12 weeks 30→60% HRR | carotid-brachial 8.7 ±0.9 | NR | 76 ±6 | 8 0% | maintain usual routine | 12 weeks | carotid-brachial 8.6 ±1.7 | NR |
| **Haynes et al. 2021** [20]* | Sedentary older adults | 62 ±7 | 18 28% | water-based walking | 29°C 1.2–1.5m | 50min 3 x/week 24 weeks 55% HHR | NR | brachial 4.4 ±2.9 | 62 ±7 | 16 25% | maintain usual routine; education | 4 non-exercise seminars over 24 weeks | NR | brachial 5.9 ±2.9 |
| **Kim et al. 2018** [35]* | Older women | 67 ±3 | 14 0% | aquarobics | 29°C 1.2m | 60min 2 x/week 16 weeks 40→70% HRR | carotid-brachial 9.3 ±0.3 | NR | 66 ±5 | 12 0% | maintain usual routine | 16 weeks | carotid-brachial 8.5 ±0.2 | NR |
| **Klonizakis and Mitropoulos 2023** [38] | Older Adults | 61 ±4 | 17 41% | swimming | NR | 45min 2-3 x/week 8 weeks Self-paced | NR | brachial 4.9 ±2.9 | 62 ±5 | 20 35% | maintain usual routine | 8 weeks | NR | brachial 4.8 ±2.2 |
| **Park et al. 2019** [18] | Peripheral artery disease | 70 ±10 | 35 %NR | water-based walking | 29°C waist-to-chest level | 60min 4 x/week 12 weeks 50→85% HR$_{max}$ | femoral-ankle 12.8 ±1.6 | NR | 71 ±8 | 37 | non-exercise activities (e.g., reading) | 60min 4 x/week 12 weeks | femoral-ankle 12.4 ±1.3 | NR |
| **Ploydang et al. 2023** [39] | Mild cognitive impairment and Type 2 diabetes | 69 ±4 | 16 31% | water-based Nordic walking | 35°C chest level | 60min 3 x/week 12 weeks 40→60% HRR | brachial-ankle 18.0 ±2.3 | brachial 4.9 ±2.2 | 69 ±5 | 17 59% | maintain usual routine | 12 weeks | brachial-ankle 17.2 ±3.0 | brachial 4.8 ±2.9 |
| **Scheer et al. 2020** [40] | Type 2 diabetes | 61 ±10 | 13 54% | water-based circuit training | 30°C NR depth | 60min 3 x/week 8 weeks 60→80% HR$_{max}$ | NR | brachial 6.1 ±2.4 | 64 ±10 | 14 57% | maintain usual routine | 8 weeks | NR | brachial 6.2 ±1.6 |
| **Scheer et al. 2023** [36]* | Coronary heart disease | 66 ±7 | 14 75% | water-based aerobic exercises | 34.5°C xiphoid level | 60min 3 x/week 12 weeks 50→80% HR$_{max}$ | NR | brachial 4.0 ±2.4 | 70 ±7 | 11 75% | maintain usual routine | 12 weeks | NR | brachial 4.2 ±2.3 |
| **Sherlock et al. 2014** [41] | Sedentary older adults | 68 ±8 | 17 31% | water-based aerobic exercise | NR | 60min 3 x/week 10 weeks 50→75% HRR | carotid-femoral 9.5 ±1.3 | NR | 69 ±8 | 16 23% | maintain usual routine | 10 weeks | carotid-femoral 8.9 ±1.1 | NR |
| **Son et al. 2024** [27] | Obese older women | 72 ±3 | 14 0% | water-based water walking | 28.5°C 1.2m | 50min 3 x/week 12 weeks 11-14 RPE | brachial-ankle 17.2 ±1.6 | NR | 71 ±5 | 12 0% | maintain usual routine | 12 weeks | brachial-ankle 16.7 ±1.7 | NR |
| **Vasić et al. 2019** [37]* | Coronary artery disease | 57 ±8 | 29 83% | water-based aerobic exercise + calisthenics | 32.8°C 1.5m | 30min 2 x/day 2 weeks 60–80% HR$_{max}$ | NR | brachial 6.6 ±2.0 | 61 ±8 | 30 20% | refrain from exercise program | 2 weeks | NR | brachial 6.4 ±2.0 |

*(Continued)*

Table 1. (Continued)

| Studies | Sample | Aquatic Exercise Group | | | | | | | Comparison Group | | | | | |
|---|---|---|---|---|---|---|---|---|---|---|---|---|---|---|
| | | Age years ±SD | n total % male | Type | Pool temperature depth | Dosage | PWV type baseline ±SD (m/s) | FMD type baseline ±SD (%) | Age years ±SD | n total % male | Type | Dosage | PWV type baseline ±SD (m/s) | FMD type baseline ±SD (%) |
| **Wong et al. 2019** [42] | Older women with hyper-tension | 75 ±3 | 52 0% | swimming | NR | 25→45min 3-4 x/week 20 weeks 60→75% HR$_{max}$ | carotid-radial 9.0 ±0.2 | NR | 74 ±4 | 48 0% | maintain usual routine | 20 weeks | carotid-radial 9.4 ±0.2 | NR |
| **Aquatic vs. Land-Based Exercise** | | | | | | | | | | | | | | |
| **Alkatan et al. 2016** [16] | Osteo-arthritis | 63 ±5 | 24 8% | swimming | 27.5°C NR depth | 20→45min 3 x/week 12 weeks, 40→70% HRR | carotid-femoral 12.9 ±1.6 | brachial 3.0 ±3.3 | 61 ±5 | 24 8% | land-based stationary bicycling | 20→45min 3 x/week 12 weeks 40→70% HRR | carotid-femoral 12.0 ±1.6 | brachial 2.9 ±2.9 |
| **Haynes et al. 2021** [20]* | Sedentary older adults | 62 ±7 | 18 39% | water-based walking | 29°C 1.2-1.5m | 50min 3 x/week 24 weeks 40→55% HHR | NR | brachial 4.4 ±2.9 | 62 ±5 | 17 18% | land-based walking | 50min 3 x/week 24 weeks 40→55% HHR | NR | brachial 5.9 ±2.9 |
| **Kim et al. 2018** [35]* | Older women | 67 ±3 | 14 0% | aquarobics | 29°C 1.2m | 60min 2 x/week 16 weeks 40→70% HRR | carotid-brachial 9.3 ±0.3 | NR | 67 ±2 | 14 0% | land-based exercise | 60min 2 x/week 16 weeks 40→70% HRR | carotid-brachial 9.3 ±0.2 | NR |
| **Lee et al. 2018** [30] | Subacute stroke | 58 ±14 | 18 43% | water-based tread-mill + physi-cal therapy | 32°C waist to knee level | 90min 5 x/week 4 weeks 30→50% HR$_{max}$ | brachial-ankle 16.0 ±3.7 | NR | 64 ±11 | 14 56% | land-based ergome-ter + physical therapy | 90min 5 x/week 4 weeks 30→50% HR$_{max}$ | brachial-ankle 17.4 ±3.7 | NR |
| **Nualnim et al. 2012** [17] | Hyper-tension | 58 ±2 | 24 42% | swimming | NR | 15→45min 3-4 x/week 12 weeks 60→75% HR$_{max}$ | NR | brachial 3.3 ±4.4 | 61 ±2 | 19 21% | land-based relaxation exercise + stretching | 15→45min 3-4 x/week 12 weeks | NR | brachial 4.8 ±3.9 |
| **Park et al. 2020** [28] | Peripheral artery disease | 60 ±9 | 28 %NR | water-based walking | 30.5°C waist to chest level | 60min 4 x/week 12 weeks 50→85% HR$_{max}$ | brachial-ankle 15.2 ±1.6 | NR | 60 ±10 | 25 %NR | land-based treadmill walking | 60min 4 x/week 12 weeks 50-85% HR$_{max}$ | brachial-ankle 15.3 ±1.4 | NR |
| **Scheer et al. 2023** [36]* | Coronary heart disease | 66 ±7 | 14 75% | water-based aerobic exercises | 34.5°C xiphoid level | 60min 3 x/week 12 weeks 50→80% HR$_{max}$ | NR | brachial 4.0 ±2.4 | 70 ±7 | 16 90% | land-based aero-bic + resis-tance exercise | 60min 3 x/week 12 weeks 50→80% HR$_{max}$ | NR | brachial 4.9 ±2.4 |
| **Suntraluck et al. 2017** [33] | Sedentary older adults with Type 2 diabetes | 60-75 (range) | 15 %NR | water-based bicycle training | 36°C hip level | 35→50min 3 x/week 12 weeks 50→70% HR$_{max}$ | brachial-ankle 17.4 ±0.6 | popliteal 3.4 ±1.9 | 60-75 (range) | 14 %NR | land-based stationary bicycling | 35→50min 3 x/week 12 weeks 50→70% HR$_{max}$ | brachial-ankle 19.1 ±3.22 | popliteal 3.4 ±1.1 |

*(Continued)*

**Table 1.** (Continued)

| Studies | Sample | Aquatic Exercise Group | | | | | | | | Comparison Group | | | | | | | |
| --- | --- | --- | --- | --- | --- | --- | --- | --- | --- | --- | --- | --- | --- | --- | --- | --- | --- |
| | | Age years ±SD | n total % male | Type | Pool temperature depth | Dosage | PWV type baseline ±SD (m/s) | FMD type baseline ±SD (%) | | Age years ±SD | n total % male | Type | Dosage | PWV type baseline ±SD (m/s) | FMD type baseline ±SD (%) | | |
| **Vasić et al. 2019 [37]*** | Coronary event | 57 ±8 | 29 83% | water-based aerobic exercise + calisthenics | 32.8°C 1.5m | 30 min 2 x/day 2 weeks 60→80% HR$_{max}$ | NR | brachial 6.6 ±2.0 | | 62 ±8 | 30 70% | land-based aerobic + calisthenics | 30 min 2 x/day 2 weeks 60→80% HR$_{max}$ | NR | brachial 5.7 ±2.3 | | |
| **Xin et al. 2024 [34]** | Sedentary men | 52 ±4 | 14 100% | HIIT water-based running | 27°C 1.2m | 40 min 3 x/week 8 weeks 35→85% HRR | carotid–femoral 14.1 ±1.8 | brachial 8.1 ±1.2 | | 53 ±6 | 13 100% | HIIT land-based running | 40 min 3 x/week 8 weeks 35→S85% HRR | carotid-femoral 14.0 ±1.5 | brachial 7.9 ±1.3 | | |

Data presented as means ± SDs unless otherwise stated. FMD: flow-mediated dilation; HIIT: high intensity interval training; HR$_{max}$: maximal heart rate; HRR: heart rate reserve; NR: not reported; PWV: pulse wave velocity; RPE: rate of perceived exertion. *Study has a non-exercise control group and land-based exercise group.

remaining five articles measured brachial-ankle PWV [27,28,30,33,39], which is considered a systemic (i.e., both central and peripheral) pathway.

## Vascular function

**Aquatic exercise versus non-exercise control.** Aquatic exercise demonstrated a significant beneficial effect on reducing overall PWV (SMD = −2.37, 95% CI: −4.46 to −0.29, P = 0.03, $I^2$ = 98%; 95% prediction interval: −9.99 to 5.25) (Fig 2a) in seven studies [18,19,27,35,39,41,42] involving 309 participants, compared with non-exercise control groups. Due to the high heterogeneity ($I^2$ = 98%), we conducted subgroup analysis based on the type of PWV measurement (systemic, peripheral, and central). In the systemic PWV subgroup (2 studies, 59 participants), heterogeneity was eliminated and a significant improvement remained (SMD = −0.59, 95% CI: −1.12 to −0.07, P = 0.03, $I^2$ = 0%). In the peripheral PWV subgroup (4 studies, 217 participants), aquatic exercise still showed a large beneficial effect, though heterogeneity remained high (SMD = −3.76, 95% CI: −6.91 to −0.60, P < 0.02, $I^2$ = 98%). In the central PWV subgroup, which included only one study (33 participants), no significant effect was found (SMD = −0.60, 95% CI: −1.30 to 0.10). For endothelial function, overall FMD (Fig 2b) increased significantly (SMD = 0.91, 95% CI: 0.39 to 1.43, P = 0.0006, $I^2$ = 68%; 95% prediction interval: −0.73 to 2.55) in aquatic groups across six studies [20,36–40] involving 215 participants. Given the substantial heterogeneity ($I^2$ = 68%), we conducted a subgroup analysis. Subgroups were defined as trials that recruited participants with a defined medical condition from those that did not specify a medical condition in their inclusion criteria. In the subgroup containing adults with a reported medical condition (4 studies, 144 participants), heterogeneity was slightly reduced and a significant improvement remained (SMD = 0.93, 95% CI: 0.36 to 1.51, P = 0.001, $I^2$ = 61%). In the subgroup with trials that did not specify a medical condition in their inclusion criteria (2 studies, 71 participants), there was no longer a significant effect and heterogeneity increased (SMD = 0.89, 95% CI: −0.53 to 2.32, P = 0.22, $I^2$ = 87%).

**Aquatic exercise versus land-based exercise.** When compared with land-based exercise, the effect of aquatic exercise on overall PWV (Fig 3a) demonstrated a non-significant intervention effect (SMD = −0.07, 95% CI: −0.34 to 0.20, P = 0.61, $I^2$ = 0%; 95% prediction interval: −0.45 to 0.31) in six studies [16,28,30,33–35] involving 217 participants. Although statistical heterogeneity was minimal and did not require subgrouping, subgroup analysis was conducted for consistency with other comparisons. In the systemic PWV subgroup (4 studies, 141 participants), results remained non-significant (SMD = −0.05, 95% CI: −0.38 to 0.29, P = 0.79, $I^2$ = 0%). The peripheral and central PWV subgroups each included one study (28 participants, 48 participants, respectively), which also showed non-significant effects (peripheral: SMD = −0.42, 95% CI: −1.17 to 0.33, P = 0.28; central: SMD 0.06, 95% CI: −0.51 to 0.62, P = 0.84). In contrast, the effect of aquatic exercise intervention on overall FMD (Fig 3b) showed a significant improvement (SMD = 0.55, 95% CI: 0.05 to 1.06, P = 0.03, $I^2$ = 75%; 95% prediction interval: −1.24 to 2.34) in seven studies [16,17,20,33,34,36,37] with 271 participants. Given the considerable heterogeneity ($I^2$ = 75%), we conducted a subgroup analysis based on study population characteristics as described earlier. In the subgroup containing adults with a reported medical condition (5 studies, 209 participants), heterogeneity was reduced and a significant improvement remained (SMD = 0.64, 95% CI: 0.23 to 1.06, P = 0.002, $I^2$ = 53%). In the subgroup with trials that did not specify a medical condition in their inclusion criteria (2 studies, 62 participants), there was no longer a significant effect and heterogeneity increased (SMD = 0.33, 95% CI: −1.58 to 2.24, P = 0.73, $I^2$ = 92%).

## Adherence and adverse events

The results of adherence and adverse events are presented in the supporting information (S1 Table). The adherence rates in aquatic exercise interventions were generally high, ranging from 79% [20] to 100% [39], with most studies reporting rates above 90% [16,17,34,38]. The average adherence rate in aquatic groups (91%) [16–18,20,28,34,38,39] was higher than in land-exercise exercise groups (89%) [16,20,28,34]. Eight of the 18 studies reported on adverse events, and none observed any events related to the aquatic exercise interventions [18,28,30,36–38,40,42]. Only one study [36] reported one event of supraventricular tachycardia in the land-based group.

## (a) Arterial stiffness: Pulse wave velocity (pre-post difference m/s)

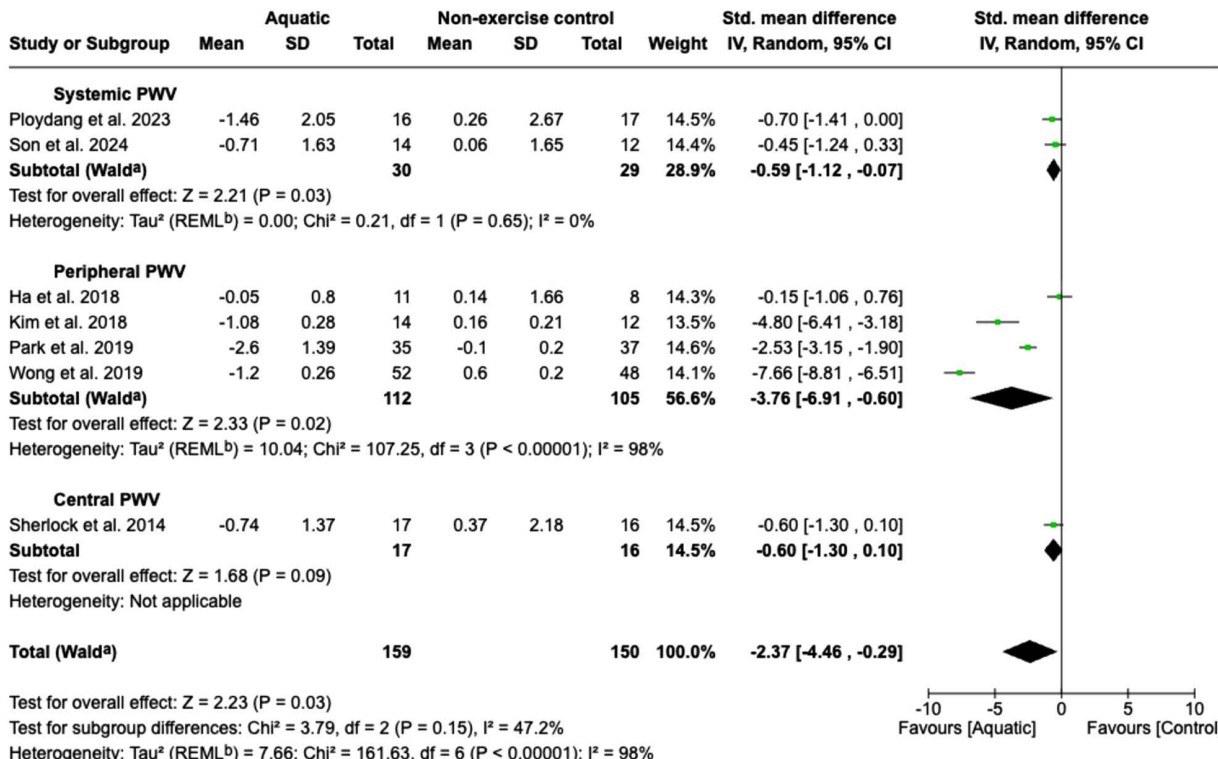

## (b) Endothelial function: Flow-mediated dilation (pre-post difference %)

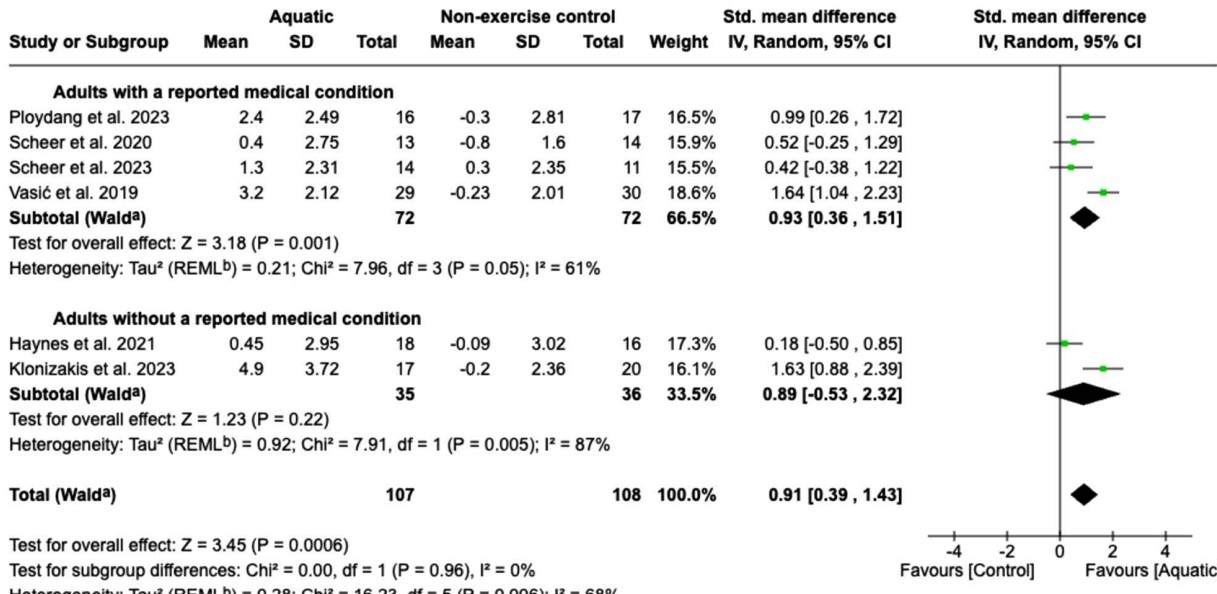

**Footnotes**

[a]CI calculated by Wald-type method.

[b]Tau² calculated by Restricted Maximum-Likelihood method.

**Fig 2. Analyses on the effects of aquatic exercise versus non-exercise control for vascular function.**

## Quality of evidence

**Risk of bias in individual studies.** Among the 18 included studies, the overall risk of bias was rated as high in 13 out of 18 studies (Fig 4a). Most studies were rated as high risk in at least one domain (Fig 4b), with common bias due to deviations from intended intervention [17,27,30,33–36,38,40,41], measurement of the outcome [16,17,27,33,34,37,39,41], and missing outcome data [17,27,35,36,40]. A more detailed explanation of the risk of bias rating, including ratings from all signalling questions, can be found in the supporting information (S2 Table).

**Certainty of evidence.** The certainty of evidence (Table 2) was downgraded for all outcomes because the proportion of information from studies at high risk of bias (Fig 4) is sufficient to affect the interpretation of the results. The outcomes assessing the effects of aquatic exercise versus non-exercise control on arterial stiffness and endothelial function were downgraded for inconsistency due to high heterogeneity ($I^2 = 98\%$ and $I^2 = 68\%$, respectively; Fig 2a). Aquatic exercise versus land-based exercise for endothelial function was also downgraded for inconsistency due to high heterogeneity across studies ($I^2 = 75\%$; Fig 3b) and because a significant portion of the confidence interval for the SMD crossed the threshold for trivial effect (SMD = 0.55, 95% CI: 0.05 to 1.06, where <0.2 is considered trivial; Fig 3b). Publication bias was not formally assessed due to fewer than 10 studies; therefore, no downgrade was applied, though the possibility of bias cannot be ruled out. Consequently, one finding had moderate certainty and three findings had low certainty (Table 2).

## Discussion

This systematic review with meta-analyses summarizes data from 18 studies to give insight into the effect of aquatic exercise on vascular function in adults. Overall, evidence indicates that aquatic exercise improves systemic and peripheral measures of arterial stiffness and endothelial function compared with non-exercise controls, while effects on central arterial stiffness remain uncertain. However, for both FMD and peripheral PWV, substantial between-study heterogeneity and wide prediction intervals indicate that the effects observed in this meta-analysis may not be consistently reproduced in new studies. Changes in arterial stiffness do not differ following aquatic or land-based exercise. Aquatic exercise may additionally lead to greater improvements in endothelial function than land-based exercise, although this conclusion remains tentative due to low-certainty evidence and substantial variation across studies.

To aid interpretation, these standardized mean differences can be expressed in approximate absolute terms. The observed reduction in arterial stiffness when comparing aquatic exercise versus non-exercise control (SMD = −2.37) equates to an approximate decrease of 1.46 m/s in PWV, while the improvement in endothelial function (SMD = 0.91) corresponds to an approximate increase of 2.35% in FMD (based on RevMan calculation of the mean difference). Both magnitudes are considered clinically meaningful and comparable to improvements typically reported after land-based aerobic training [45]. These values represent rough approximations based on back-transformation of pooled standardized effects, rather than pooled mean differences from the included trials, and should therefore be interpreted cautiously.

The dosage of the aquatic exercise programs in the included studies averaged 50 minutes, 3 times a week for 11 weeks and consisted predominantly of moderate-to-vigorous intensity aerobic exercise. This is similar to the dosage and type (i.e., aerobic) of land-based exercise that is known to improve PWV and FMD in adults [45]. The consistency between effective aquatic and land-based exercise prescriptions suggests that established aerobic exercise principles translate well to the aquatic environment, potentially offering similar cardiovascular benefits through comparable training stimuli.

The greater endothelial function improvements observed with aquatic exercise compared with land-based exercise in this meta-analysis should be interpreted cautiously, given the overall low certainty of evidence and the limited number of contributing studies. However, this pattern may be partially explained by the unique physiological environment created

## (a) Arterial stiffness: Pulse wave velocity (pre-post difference m/s)

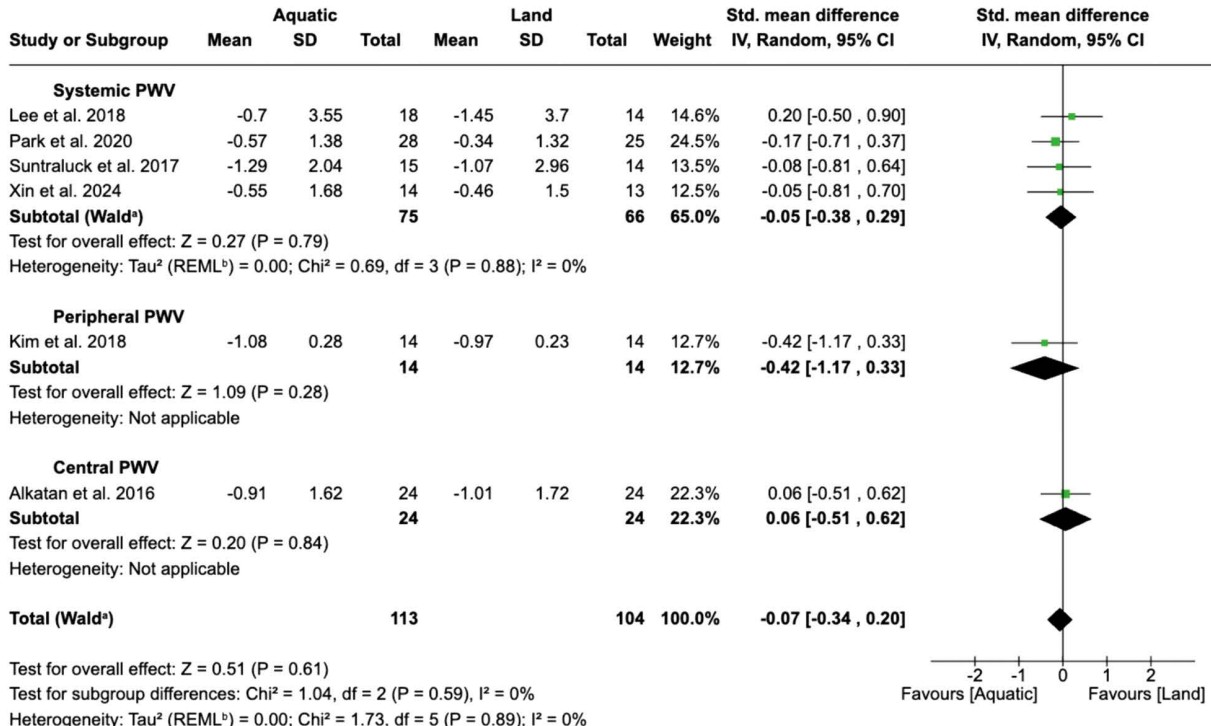

## (b) Endothelial function: Flow-mediated dilation (pre-post difference %)

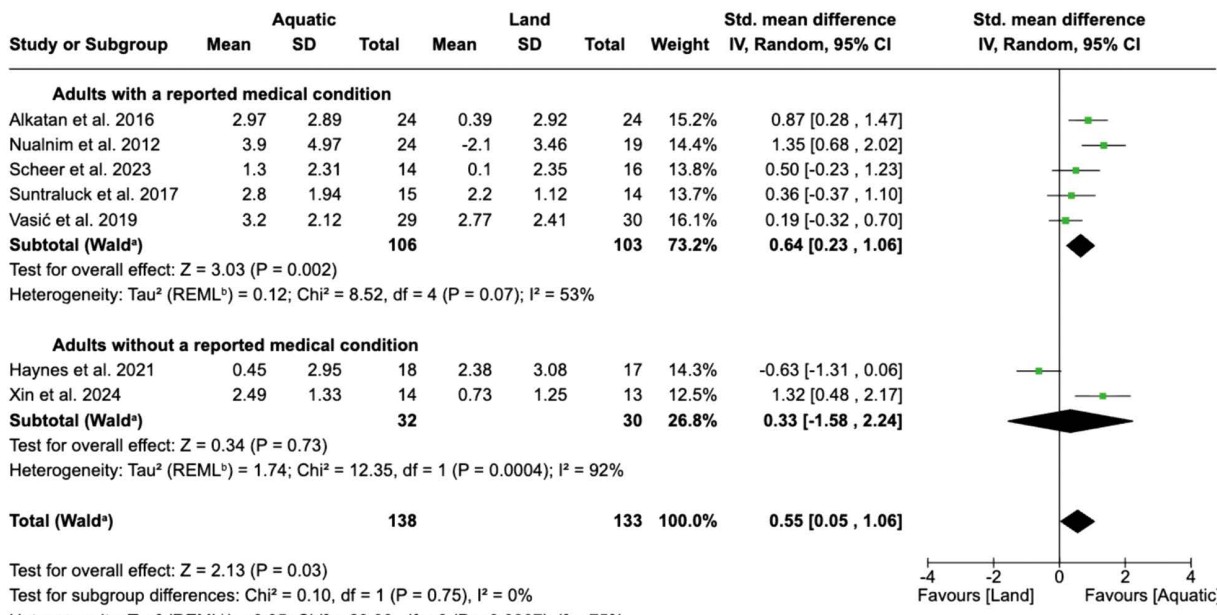

**Footnotes**

[a]CI calculated by Wald-type method.

[b]Tau² calculated by Restricted Maximum-Likelihood method.

**Fig 3. Analyses on the effects of aquatic exercise versus land-based exercise for vascular function.**

## (a) Risk of bias for each included trial

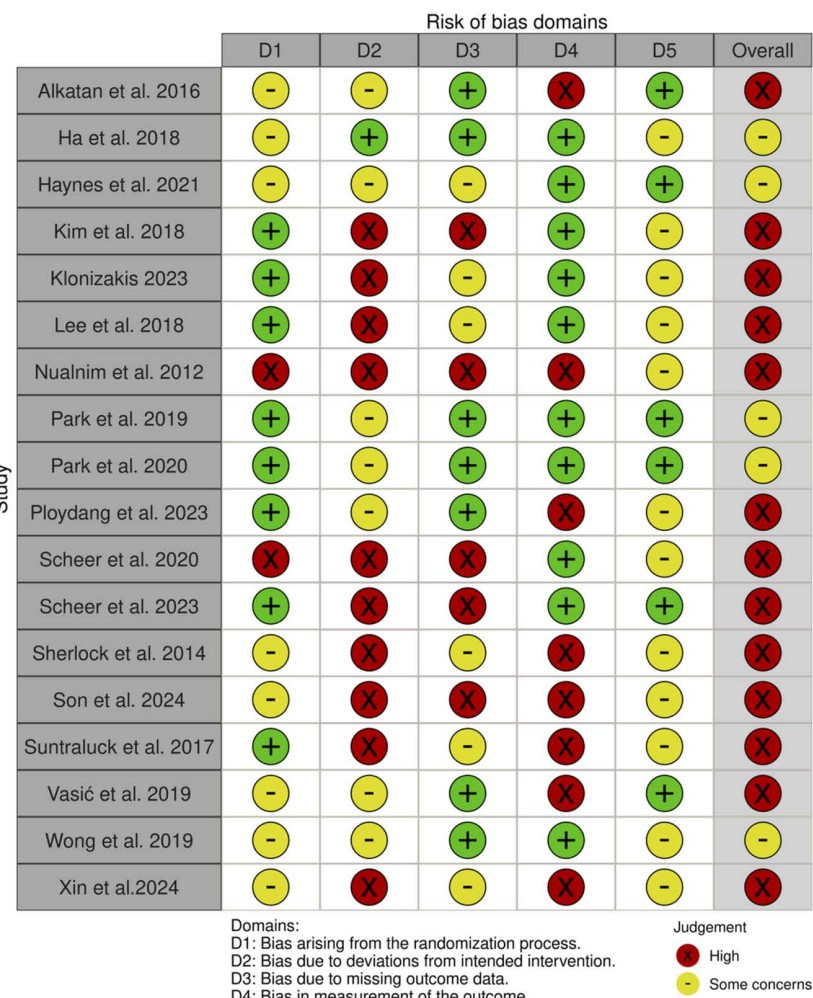

Domains:
D1: Bias arising from the randomization process.
D2: Bias due to deviations from intended intervention.
D3: Bias due to missing outcome data.
D4: Bias in measurement of the outcome.
D5: Bias in selection of the reported result.

Judgement
X High
- Some concerns
+ Low

## (b) Summary of risk of bias for all trials

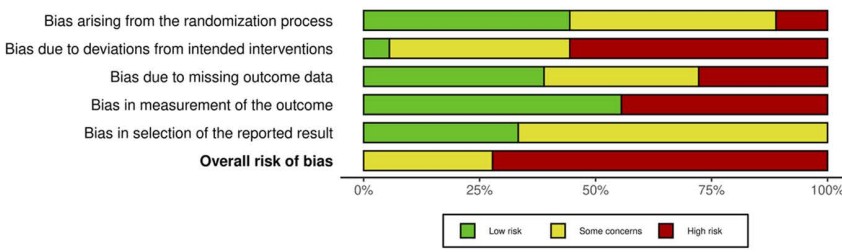

**Fig 4. Risk of bias assessment.**

**Table 2. Summary: Aquatic exercise effects on vascular function.**

| Outcomes | Standardized effects (95% CI) | № of participants (studies) | Certainty of the evidence (GRADE) | Comments |
|---|---|---|---|---|
| **Aquatic vs Non-Exercise Control Group** | | | | |
| Change in arterial stiffness | SMD 2.37 SD lower (4.46 lower to 0.29 lower) | 309 (7 RCTs) | ⊕⊕○○ Low[a,b] | Aquatic exercise **may improve arterial stiffness** as seen with a post-intervention reduction in PWV. Most evidence reflects systemic and peripheral PWV; effects on central PWV remain uncertain. |
| Change in endo-thelial function | SMD 0.91 SD higher (0.39 higher to 1.43 higher) | 215 (6 RCTs) | ⊕⊕○○ Low[a,b] | Aquatic exercise **may improve endothelial function** as seen with a post-intervention increase in FMD. |
| **Aquatic vs Land-Based Exercise** | | | | |
| Change in arterial stiffness | SMD 0.07 SD lower (0.34 lower to 0.20 higher) | 217 (6 RCTs) | ⊕⊕⊕○ Moderate[a] | Aquatic exercise **likely results in little to no difference in the change in arterial stiffness relative to land-based** exercise. |
| Change in endo-thelial function | SMD 0.55 SD higher (0.05 higher to 1.06 higher) | 271 (7 RCTs) | ⊕⊕○○ Low[a,c] | Aquatic exercise **may improve endothelial function more than land-based exercise** as seen with greater post-intervention increase in FMD. |

CI: confidence interval; FMD: flow-mediated dilation; PWV: pulse wave velocity; RCT: randomized controlled trial; SD: standard deviation; SMD: standardized mean difference.

***Explanations for Working Group Grades of Evidence (GRADE)***

[a]Downgrade due to high overall risk of bias ratings for the majority of trials; [b]Downgrade due to considerable heterogeneity across trials. [c]Downgrade due to considerable heterogeneity across trials and the 95%CI of the SMD spans from a trivial to large effect.

by water immersion. The hydrostatic pressure exerted by water creates a pressure gradient that enhances venous return and increases central blood volume, leading to elevated cardiac output and enhanced blood flow throughout the vascular system [14]. This increased blood flow enhances shear stress on the vascular endothelium, which serves as a stimulus for endothelial adaptations [7]. The magnitude of shear stress experienced during aquatic exercise is known to exceed that of equivalent land-based activities due to the combined effects of hydrostatic pressure-induced increases in venous return and the resistance provided by water movement during exercise, potentially enhancing shear stress exposure across a broader vascular network [11,46]. However, these mechanistic interpretations remain speculative and were not directly tested in the included randomized controlled trials. Mechanistic evidence from a randomized crossover trial has shown that exercise during water immersion at 32 °C can increase retrograde shear stress in young men, a hemodynamic pattern generally considered unfavorable for endothelial adaptations. In contrast, immersion at 38 °C elicited predominantly antegrade shear stress, which is typically associated with favorable vascular outcomes. Importantly, this work examined immersion only to the umbilicus and the acute effects of a single 30-minute exercise bout [14]. By contrast, studies included in our review predominantly enrolled older adults and individuals with cardiovascular or metabolic conditions, with immersion depths ranging from waist to chest level (most at chest), and involved habitual training interventions over several weeks to months, with sessions typically lasting 45–60 minutes. Reported pool temperatures ranged from 27 °C to 36 °C, while several studies did not report water temperature at all. These differences in immersion depth, water temperature, population, and training duration may help explain variability in outcomes across studies. Although acute retrograde shear during aquatic exercise has been observed in mechanistic studies, repeated training and systemic adaptations may mitigate such short-term effects. Future research should integrate systematic reporting of immersion depth, water temperature, and participant characteristics, along with direct hemodynamic measurements, to clarify the conditions under which aquatic exercise most effectively promotes vascular health.

Beyond these mechanistic considerations, participant characteristics appear to influence outcomes. Subgroup analysis indicated that trials enrolling adults with a reported medical condition demonstrated significant improvements in FMD, whereas trials including populations without a reported medical condition did not show significant changes. This pattern

suggests that individuals with existing vascular or metabolic risk factors may derive greater benefit from aquatic exercise, possibly due to lower baseline endothelial function and a wider margin for physiological adaptation. Because this sub-group analysis was exploratory rather than pre-registered, its findings should be interpreted with caution. Other factors such as water temperature, immersion depth, participant age, and exercise modality could not be evaluated due to insufficient or inconsistent reporting across studies, leaving possible confounders unaddressed. This complexity underscores the need to consider both individual and environmental factors when optimizing aquatic exercise interventions for vascular health.

Aquatic exercise provides a valuable alternative for individuals who may face barriers to land-based physical activity. Common barriers to exercise participation include physical symptoms and fear of pain or injury [47]. The buoyant properties of water address many of these concerns by reducing joint loading and impact forces, making exercise more accessible for people with arthritis, joint pain, mobility limitations, or those recovering from injury [48–50]. In people with limited mobility and weakness, water immersion decreases joint loading and supports movements that may be difficult to perform on land, making exercise safe, enjoyable, and feasible [51]. Consistent with these theoretical advantages, our findings demonstrate good adherence with aquatic exercise, with eight studies reporting adherence rates averaging 91% [16–18,20,28,34,38,39]. However, when comparing adherence between aquatic and land-based exercise [16,20,28,34], no clear advantage emerged for either modality. This may be due to the small number of comparative studies, different exercise protocols, and varying program durations. For instance, in a six-month study, aquatic walking achieved 79% adherence compared to 81% for outdoor land-walking [20], while a three-month study showed 88% adherence for aquatic walking exercise versus 81% for land-based treadmill walking exercise [28]. One could assume lower adherence to a six month program as individuals may lose the initial enthusiasm for study participation after such a long duration. On the other hand, it is possible in the three-month program that the water walking exercise may allow for more social engagement and less physical discomfort than a land-based treadmill program, which could be a reason for the lower adherence in the land-based exercise program. While we do not have direct qualitative data to confirm this, these contextual factors may help explain variation in adherence across environments.

Safety is another important consideration when recommending exercise modalities, particularly for populations with health conditions or mobility limitations. Among the studies that reported safety outcomes (8 of 18 studies), aquatic exercise appears to be well-tolerated, with no adverse events reported during the water-based sessions [18,28,30,36–38,40,42], while one adverse event (an episode of supraventricular tachycardia requiring medication) [36] was reported during land-based training. Given the observed improvement in endothelial function with aquatic exercise, water-based programs may offer both accessibility advantages and enhanced vascular function benefits for diverse populations who face barriers to traditional land-based exercise.

The various PWV measurement pathways in our review may explain some of the variability in our PWV results when comparing aquatic exercise to non-exercise controls. It is known that central and peripheral artery stiffness respond differently to exercise training [52,53], a pattern reflected in our subgroup analyses. To explore the substantial heterogeneity observed in the overall PWV meta-analysis ($I^2$=98%; Fig 2a), we stratified studies by arterial measurement type (systemic, peripheral, and central). In the systemic subgroup, heterogeneity was eliminated ($I^2$=0%; Fig 2a) and the beneficial effect remained significant (SMD=−0.59, 95% CI −1.12 to −0.07). In contrast, the peripheral subgroup retained high heterogeneity ($I^2$=98%; Fig 2a) despite a strong pooled effect estimate (SMD=−3.76, 95% CI −6.91 to −0.60). This suggests measurement location may be a key contributor to heterogeneity, consistent with physiological differences in arterial responses across systemic, peripheral and central vessels.

Despite these promising findings for aquatic exercise on vascular function, several limitations should be acknowledged. This systematic review included studies that primarily reported participant sex, with limited information on other demographic factors, and we acknowledge this as a limitation in assessing equity, diversity, and inclusion across the evidence base. Relatively high statistical heterogeneity across some outcomes limited our ability to draw definitive conclusions.

Although subgroup analyses clarified some of this variation, residual heterogeneity may stem from differences in participant characteristics (e.g., age, health status, and comorbidities), intervention types (e.g., aquatic walking vs. swimming), and exercise protocols (e.g., supervision, type, intensity, and duration). However, concerns about population-related heterogeneity are somewhat mitigated by land-based exercise research demonstrating similar PWV and FMD improvements across diverse populations, including older adults and those with conditions such as diabetes, hypertension [45], and coronary heart disease [54]. Risk of bias contributed to the reduced level of certainty, with 13 of 18 studies showing high risk mainly due to attrition bias and lack of outcome assessor blinding. These methodological limitations (high risk of bias and heterogeneity) may have inflated the observed effects, and we highlight this as a key limitation affecting confidence in the findings. Some biases are inherent to exercise interventions. Participant and personnel blinding was not practically feasible for any study, and attrition bias is particularly challenging given the sustained commitment required over extended periods. However, some identified biases offer insight for future research. The lack of assessor blinding is concerning for FMD measurements, as this outcome can be influenced by the assessor's knowledge of the intervention group, whereas this is less of an issue with the semi-automatic measure of PWV. Additionally, many biases resulted from loss to follow-up, highlighting the importance of developing strategies to improve retention.

While acknowledging these considerations, our systematic review provides several methodological advantages and novel insights that advance understanding of aquatic exercise and vascular function. Most notably, this represents the first systematic review to exclusively examine randomized controlled trials of chronic aquatic exercise on vascular function, providing enhanced methodological rigor compared to previous work in this area. By restricting our analysis to chronic exercise training and excluding acute exercise studies, we have eliminated a major source of heterogeneity that has complicated previous land-based exercise systematic reviews, which conflate immediate post-exercise responses with sustained training adaptations. This methodological distinction is critical, as acute exercise responses may not accurately reflect the long-term vascular benefits achievable through sustained training programs.

## Conclusion

Our findings indicate that aquatic exercise programs – averaging 50 minutes, three times per week for 11 weeks at a moderate-to-vigorous intensity – improve endothelial function as well as systemic and peripheral arterial stiffness compared with non-exercise controls, while effects on central arterial stiffness remain uncertain. Aquatic exercise likely results in little to no difference in arterial stiffness compared with land-based exercise and may be more effective for improving endothelial function, although this latter finding is based on low-certainty evidence and is limited by high heterogeneity across included studies. These programs were well-tolerated, with high adherence and no reported adverse effects. Collectively, these findings suggest that aquatic exercise is a promising alternative to land-based training, and can be particularly valuable for individuals with barriers to traditional land-based exercise. However, future research should focus on well-designed randomized controlled trials with larger sample sizes, blinded assessors, and strategies for participant retention, in order to strengthen the evidence base and help refine aquatic exercise prescriptions for vascular function benefits.

## Supporting information

**S1 File. Database search terms.**
(PDF)

**S1 Table. Adherence and adverse events.**
(PDF)

**S2 Table. Revised Cochrane risk of bias tool for randomized trials (RoB 2) – Signalling question for included studies.**
(PDF)

## Author contributions

**Conceptualization:** Emily Dunlap, Manny M.Y. Kwok, Billy C. L. So, Hirofumi Tanaka.

**Data curation:** Emily Dunlap, Yanbing Zhou, Manny M.Y. Kwok.

**Formal analysis:** Emily Dunlap, Hirofumi Tanaka.

**Methodology:** Emily Dunlap, Yanbing Zhou, Manny M.Y. Kwok, Billy C. L. So, Hirofumi Tanaka.

**Project administration:** Emily Dunlap, Hirofumi Tanaka.

**Software:** Emily Dunlap.

**Supervision:** Emily Dunlap, Hirofumi Tanaka.

**Validation:** Emily Dunlap, Yanbing Zhou, Hirofumi Tanaka.

**Visualization:** Emily Dunlap, Hirofumi Tanaka.

**Writing – original draft:** Emily Dunlap, Yanbing Zhou, Manny M.Y. Kwok, Hirofumi Tanaka.

**Writing – review & editing:** Emily Dunlap, Yanbing Zhou, Manny M.Y. Kwok, Billy C. L. So, Hirofumi Tanaka.

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
