## [Decision Letter · Decision Letter 0]

12 Aug 2025

Dear Dr. Dunlap,

Thank you for submitting your manuscript to PLOS ONE. After careful consideration, we feel that it has merit but does not fully meet PLOS ONE’s publication criteria as it currently stands. Therefore, we invite you to submit a revised version of the manuscript that addresses the points raised during the review process.

We look forward to receiving your revised manuscript.

Kind regards,

Hidetaka Hamasaki

Academic Editor

PLOS ONE

Journal Requirements:

2. In the online submission form, you indicated that the data underlying the results presented in the study are available from the corresponding author (edunlap@utexas.edu ).

Reviewers' comments:

Reviewer's Responses to Questions

**Comments to the Author**

1. Is the manuscript technically sound, and do the data support the conclusions?

Reviewer #1: Partly

Reviewer #2: Yes

Reviewer #3: Partly

2. Has the statistical analysis been performed appropriately and rigorously?

Reviewer #1: Yes

Reviewer #2: Yes

Reviewer #3: Yes

3. Have the authors made all data underlying the findings in their manuscript fully available?

Reviewer #1: Yes

Reviewer #2: Yes

Reviewer #3: Yes

4. Is the manuscript presented in an intelligible fashion and written in standard English?

Reviewer #1: Yes

Reviewer #2: Yes

Reviewer #3: Yes

Reviewer #1: This is a well-conducted systematic review of aquatic exercise interventions on vascular function. The authors address the clinically important topic, but the manuscript would benefit from a more nuanced interpretation of the findings. The review provides moderate-certainty evidence that aquatic exercise improves vascular function, particularly endothelial function, more than no exercise and possibly more than land-based exercise. However, conclusions should be toned down as there are methodological limitations, high heterogeneity, and risk of bias in the included studies. However, this manuscript contributes valuable insight into how aquatic exercise may influence vascular health, particularly in older adults or clinical populations.

MAJOR COMMENTS:

The pooled analysis for PWV versus non-exercise has an I2 of 98%, which is extremely high. Moderate GRADE ratings were given even when risk of bias and heterogeneity were both high. The justification for not downgrading for inconsistency is debatable. The argument that sensitivity analysis "rescues" the finding is not sufficient to ignore heterogeneity entirely.

While a sensitivity analysis helps explain some of this, the clinical and methodological sources of heterogeneity (e.g., water temperature, exercise modality, protocol duration) need clearer exploration. Does the heterogeneity reduce the confidence in the generalizability of the findings? Maybe a subgroup analysis is needed..

The impact of measurement variability on effect sizes and heterogeneity needs more discussion. Studies using peripheral versus central pathways are not equivalent. Did the authors adjust the meta-analysis to account for which segment of the vasculature was measured? For example, did they analyze central PWV separately from peripheral PWV? Or did they combine them all together?

The authors should explore more granular sources of heterogeneity, such as population characteristics, exercise modality (e.g., aqua-aerobics vs. swimming), and vascular segment measured. Consider re-running sensitivity analyses excluding high-risk studies altogether to see if main conclusions still hold.

The conclusions rely heavily on studies with high risk of bias, yet this issue is underemphasized in the discussion. The authors should more explicitly discuss how these limitations might have distorted the findings, particularly for FMD. Tone down conclusions of “superior” effect and emphasize suggestive rather than definitive evidence, in line with GRADE and CI boundaries.

MINOR COMMENTS:

In the abstract, try to include exact values: SMDs, CIs, I² for PWV and FMD outcomes, and clearly highlight the main conclusion.

In Table 1, water depth is missing for several interventions. Please clarify if the information was not reported by the study or if it was omitted here.

For consistency and clarity, also provide mean ± SD (if available) for age so readers can compare across studies. A range alone is less informative.

Page 20: “This increased blood flow generates greater flow throughout the vascular system.”

This sentence is redundant. Consider revising to: “This increased blood flow enhances shear stress throughout the vascular system.”

Page 21: “This mechanistic explanation is consistent with our findings of superior endothelial function...”

Provide a specific citation to support the mechanistic claim or refer back to a particular figure/table in your analysis.

Page 22: “The water walking exercise may allow for more social engagement...”

Consider acknowledging this as a hypothesis or anecdotal observation unless you have qualitative data to support it.

Reviewer #2: The manuscript has strong potential, but it requires improvements in multiple sections, particularly in clarity, transparency of reporting, and adherence to systematic review best practices.

Abstract

Include more specific details about the search period (e.g., “from database inception to April 16, 2025”).

The conclusion should mention the heterogeneity observed, as this affects the interpretation of the meta-analysis results.

Methods

Line 138-147: The inclusion criteria should be more detailed. Specify the participants characteristics and types of interventions explicitly.

Line 179: Provide more detail on the meta-analysis approach, particularly the rationale for using the random-effects model.

Results

Page 15, first paragraph: Describe the characteristics of the included studies in more detail. For example, include information on the sample, sample size range, and the settings of the interventions.

Page 15, first paragraph: Please rephrase the sentence ‘Water walking18,20,27,28,30,39 and combined walking-aerobic exercises,19,35–37,40,41 with six studies each’ for clarity.

Page 16, second paragraph: Address the potential impact of heterogeneity. The high I² values reported (e.g., I²=98%) indicate substantial heterogeneity, which should be explored further in the text.

Discussion

Provide a more detailed discussion of the potential sources of heterogeneity observed in the meta-analysis. Discuss factors such as differences in sample characteristics, intervention types, and dosage.

Suggest clearer implications for clinical practice and future research. For example, recommend specific areas where further high-quality RCTs are needed.

Conclusion

Emphasize the need for better-designed RCTs that address the limitations identified in this review, particularly the issue of inadequate demographic factors and heterogeneity.

Figures and Tables

Strengths: The figures and tables are informative and well-organized.

Improvements: Ensure all acronyms are defined in the table legends in Table 2 (e.g., SMD, PWV, FMD, RCT). The forest plots should include a clearer indication of the confidence intervals and heterogeneity statistics (I² values).

References

Ensure all references are formatted consistently according to PLOS One’s citation style, particularly regarding journal titles.

Reviewer #3: Benefits in vascular function (endothelial function and arterial stiffness) with aquatic exercise compared to land-based exercise have been reported previously. The current manuscript confirms those benefits but, as written, does not present a clearly novel finding for long-term vascular health in older adults: when exercise is performed consistently, aquatic and land-based modalities appear similarly effective.

To add value and clarify effect modifiers, the authors should consider subgroup analyses stratified by age, sex, and disease status (see recommended subgroup scheme below). These comparisons would help reveal whether aquatic exercise confers different benefits in specific populations (e.g., older women, younger men with diabetes) and would strengthen the manuscript’s contribution.

Finally, the Risk-of-Bias table is not sufficiently clear. The authors should expand and clarify the RoB presentation so readers can assess study quality at a glance.

Recommended subgrouping scheme (for analysis & figures)

Perform subgroup analyses for each outcome (FMD, pulse wave velocity/arterial stiffness, NO metabolites, etc.) using the following strata:

Older men — healthy (aquatic vs land vs control)

Older men — disease (e.g., diabetes, established CVD)

Younger men — healthy (aquatic vs land vs control)

Younger men — disease (e.g., diabetes, established CVD)

Older women — healthy (aquatic vs land vs control)

Older women — disease (e.g., diabetes, established CVD)

Younger women — healthy (aquatic vs land vs control)

Younger women — disease (e.g., diabetes, established CVD)

**Do you want your identity to be public for this peer review?** For information about this choice, including consent withdrawal, please see our Privacy Policy

Reviewer #1: **Yes: ** Shane J.T. Balthazaar

Reviewer #2: No

Reviewer #3: No

---

## [Author Response · Author response to Decision Letter 1]

8 Sep 2025

A point-by-point response to reviewers can be found in the attached file labeled 'Response to reviewers'.

---

## [Decision Letter · Decision Letter 1]

1 Oct 2025

Dear Dr. Dunlap,

Thank you for submitting your manuscript to PLOS ONE. After careful consideration, we feel that it has merit but does not fully meet PLOS ONE’s publication criteria as it currently stands. Therefore, we invite you to submit a revised version of the manuscript that addresses the points raised during the review process.

We look forward to receiving your revised manuscript.

Kind regards,

Hidetaka Hamasaki

Academic Editor

PLOS ONE

Journal Requirements:

Reviewers' comments:

Reviewer's Responses to Questions

**Comments to the Author**

Reviewer #1: (No Response)

Reviewer #2: All comments have been addressed

2. Is the manuscript technically sound, and do the data support the conclusions?

Reviewer #1: Partly

Reviewer #2: Yes

3. Has the statistical analysis been performed appropriately and rigorously?

Reviewer #1: No

Reviewer #2: Yes

4. Have the authors made all data underlying the findings in their manuscript fully available?

Reviewer #1: Yes

Reviewer #2: Yes

5. Is the manuscript presented in an intelligible fashion and written in standard English?

Reviewer #1: Yes

Reviewer #2: Yes

Reviewer #1: The authors have strengthened the manuscript since the prior round, especially adjusting conclusions around FMD in line with GRADE. However, some comments still remain:

The current “central/systemic” subgroup conflates cfPWV (central) with baPWV (systemic). These are related but distinct constructs and should not be pooled together. Please re-run and report PWV in three strata: central (cfPWV), systemic (baPWV), and peripheral segmental (e.g., crPWV, cbPWV, faPWV).

The SMD for peripheral PWV stands out (-3.76)? Can we have another look at the SDs from each study – one study can make the effect size inflated. Were medians or standard errors converted correctly? Maybe a “leave-one-out” analysis is needed here… especially with high heterogeneity…

Why use SMD for PWV? If the SDs are very different between studies, the SMD values can become misleading… maybe use mean differences in m/s

It will be helpful to explicitly state that the 95%CI for aquatic vs land spans from trivial to moderate, justifying the low-certainty grade

Let the reader know that the subgroup analysis with the medical condition group vs healthy is a between study comparison… there were differences in water temperature and depth, etc.

Testing for publication bias is only meaningful with at least 10 studies…

Reviewer #2: (No Response)

**Do you want your identity to be public for this peer review?** For information about this choice, including consent withdrawal, please see our Privacy Policy

Reviewer #1: **Yes: ** Shane J.T. Balthazaar

Reviewer #2: **Yes: ** Ulric Sena Abonie

---

## [Author Response · Author response to Decision Letter 2]

7 Oct 2025

Please see the uploaded document titled "Response for Reviewers" for point-by-point response to reviewers and editor.

---

## [Decision Letter · Decision Letter 2]

19 Oct 2025

Dear Dr. Dunlap,

Thank you for submitting your manuscript to PLOS ONE. After careful consideration, we feel that it has merit but does not fully meet PLOS ONE’s publication criteria as it currently stands. Therefore, we invite you to submit a revised version of the manuscript that addresses the points raised during the review process.

We look forward to receiving your revised manuscript.

Kind regards,

Hidetaka Hamasaki

Academic Editor

PLOS ONE

Journal Requirements:

Reviewers' comments:

Reviewer's Responses to Questions

**Comments to the Author**

Reviewer #1: (No Response)

2. Is the manuscript technically sound, and do the data support the conclusions?

Reviewer #1: Partly

3. Has the statistical analysis been performed appropriately and rigorously?

Reviewer #1: Yes

4. Have the authors made all data underlying the findings in their manuscript fully available?

Reviewer #1: Yes

5. Is the manuscript presented in an intelligible fashion and written in standard English?

Reviewer #1: Yes

Reviewer #1: There are clear improvements to the manuscript, but a few reporting issues and some things with interpretation still remain.

Since the central PWV subgroup had one study with a non-significant effect, please revise the conclusion and abstract to specify that it was systemic PWV that improved vs non-exercise and central PWV remains uncertain.

Prediction intervals need to be added to all main meta-analyses. Consider Hartung-Knapp sensitivity for high heterogeneity as well.

Fair point with SMDs, but they are still hard to interpret clinically. Could the authors please add a translation of SMDs into approximate absolute units (% for FMD, m/s for PWV). This could simply be in the discussion.

Was there a difference in FMD protocols in these studies? (arterial site, cuff position, occlusion duration)

SMD -0.07 does not mean "similar effects" - more neutral phrasing is needed here (e.g., "did not differ")

Adverse events were mostly not reported - this has to be better reported

**Do you want your identity to be public for this peer review?** For information about this choice, including consent withdrawal, please see our Privacy Policy

Reviewer #1: **Yes: ** Shane J.T. Balthazaar

---

## [Author Response · Author response to Decision Letter 3]

30 Oct 2025

Please see uploaded document - Response to reviewers.

---

## [Decision Letter · Decision Letter 3]

12 Nov 2025

Dear Dr. Dunlap,

Thank you for submitting your manuscript to PLOS ONE. After careful consideration, we feel that it has merit but does not fully meet PLOS ONE’s publication criteria as it currently stands. Therefore, we invite you to submit a revised version of the manuscript that addresses the points raised during the review process.

We look forward to receiving your revised manuscript.

Kind regards,

Hidetaka Hamasaki

Academic Editor

PLOS ONE

Journal Requirements:

Reviewers' comments:

Reviewer's Responses to Questions

**Comments to the Author**

Reviewer #1: (No Response)

2. Is the manuscript technically sound, and do the data support the conclusions?

Reviewer #1: Yes

3. Has the statistical analysis been performed appropriately and rigorously?

Reviewer #1: Yes

4. Have the authors made all data underlying the findings in their manuscript fully available?

Reviewer #1: Yes

5. Is the manuscript presented in an intelligible fashion and written in standard English?

Reviewer #1: Yes

Reviewer #1: Some highly heterogeneous results (e.g., FMD aquatic vs land-based; peripheral PWV) appear in the abstract and conclusion without an explicit statement that heterogeneity limits generalizability.

Abstract (lines 46–47):

“Aquatic exercise may provide greater improvement in endothelial function than land-based exercise, though this is supported by low-certainty evidence.”

Please add at the end:

“…though this is supported by low-certainty evidence, and substantial heterogeneity limits confidence in the generalizability of this finding.”

Discussion (lines 434–437): Please add “However, for both FMD and peripheral PWV, substantial between-study heterogeneity and wide prediction intervals indicate that the effects observed in this meta-analysis may not be consistently reproduced in new, similar studies.”

Conclusion (lines 585–588):

Please change “…and may be more effective for improving endothelial function, although this latter finding is based on low-certainty evidence.” to “…and may be more effective for improving endothelial function, although this finding is based on low-certainty evidence and is limited by high heterogeneity across included studies.”

Back-translated SMD... absolute units should be described as rough approximations, not pooled mean differences.

Discussion lines 440–447, where you introduce the approximate “clinical translation.”

After the sentence ending “…comparable to improvements typically reported after land-based aerobic training [45].” add:

“These values represent rough approximations based on back-transformation of pooled standardized effects, rather than pooled mean differences from the included trials, and should therefore be interpreted cautiously.”

Subgroup (medical vs non-medical) was not pre-registered and other moderators could plausibly influence effects but were unanalyzable.

Methods (after describing subgroup by medical condition), please add: “This subgroup was not pre-specified in the PROSPERO registration but was conducted post-hoc to explore potential sources of heterogeneity observed in the primary analyses.”

Discussion (lines 487–497), after the sentence ending “…wider margin for physiological adaptation.” please add: “Because this subgroup analysis was exploratory rather than pre-registered, its findings should be interpreted with caution. Other factors such as water temperature, immersion depth, participant age, and exercise modality could not be evaluated due to insufficient or inconsistent reporting across studies, leaving possible confounders unaddressed.”

Some wording in Abstract and Discussion is still slightly stronger than warranted by low-certainty GRADE ratings (though the authors have made some adjustments since the last submission):

Abstract (lines 40–41): “Aquatic exercise improved outcomes versus non-exercise controls…” should be “Aquatic exercise may improve outcomes versus non-exercise controls…”

Discussion (lines 436–437): “Aquatic exercise may additionally lead to greater improvements in endothelial function than land-based exercise…” should be “Aquatic exercise may additionally lead to greater improvements in endothelial function than land-based exercise, although this conclusion remains tentative due to low-certainty evidence and substantial variation across studies.”

Discussion (lines 455–458): Don't imply causal certainty when describing hydrostatic pressure.. maybe say “…although these mechanistic interpretations remain speculative and were not directly tested in the included trials.”

Conclusion (lines 585–588): “Aquatic exercise likely results in little to no difference in arterial stiffness compared with land-based exercise and may be more effective for improving endothelial function, although this latter finding is based on low-certainty evidence and high heterogeneity.”

**Do you want your identity to be public for this peer review?** For information about this choice, including consent withdrawal, please see our Privacy Policy

Reviewer #1: **Yes: ** Shane J.T. Balthazaar

---

## [Author Response · Author response to Decision Letter 4]

13 Nov 2025

Please see the rebuttal letter that responds to each point raised by the reviewer.

---

## [Decision Letter · Decision Letter 4]

1 Dec 2025

Effects of aquatic exercise on arterial stiffness and endothelial function in adults: a systematic review and meta analyses

PONE-D-25-37368R4

Dear Dr. Dunlap,

We’re pleased to inform you that your manuscript has been judged scientifically suitable for publication and will be formally accepted for publication once it meets all outstanding technical requirements.

Kind regards,

Hidetaka Hamasaki

Academic Editor

PLOS ONE

Additional Editor Comments (optional):

Reviewers' comments:

Reviewer's Responses to Questions

**Comments to the Author**

Reviewer #1: All comments have been addressed

2. Is the manuscript technically sound, and do the data support the conclusions?

Reviewer #1: Yes

3. Has the statistical analysis been performed appropriately and rigorously?

Reviewer #1: Yes

4. Have the authors made all data underlying the findings in their manuscript fully available?

Reviewer #1: Yes

5. Is the manuscript presented in an intelligible fashion and written in standard English?

Reviewer #1: Yes

Reviewer #1: Thank you for addressing my comments over the last few months. It is a good addition to the literature.

**Do you want your identity to be public for this peer review?** For information about this choice, including consent withdrawal, please see our Privacy Policy

Reviewer #1: **Yes: ** Shane J.T. Balthazaar

---

## [Editor Report · Acceptance letter]

PONE-D-25-37368R4

PLOS One

Dear Dr. Dunlap,

I'm pleased to inform you that your manuscript has been deemed suitable for publication in PLOS One. Congratulations! Your manuscript is now being handed over to our production team.

Kind regards,

on behalf of

Dr. Hidetaka Hamasaki

Academic Editor

PLOS One